# Fluid Flow Mechanical Stimulation-Assisted Cartridge Device for the Osteogenic Differentiation of Human Mesenchymal Stem Cells

**DOI:** 10.3390/mi12080927

**Published:** 2021-08-03

**Authors:** Ki-Taek Lim, Dinesh-K. Patel, Sayan-Deb Dutta, Keya Ganguly

**Affiliations:** 1Department of Biosystems Engineering, Institute of Forest Science, Kangwon National University, Chuncheon 24341, Korea; dineshbhud10@gmail.com (D.-K.P.); sayan91dutta@gmail.com (S.-D.D.); gkeya14@gmail.com (K.G.); 2Biomechagen Co., Ltd., Chuncheon 24341, Korea

**Keywords:** cartridge device, perfusion, mesenchymal stem cells, gene expression

## Abstract

Human mesenchymal stem cells (hMSCs) have the potential to differentiate into different types of mesodermal tissues. In vitro proliferation and differentiation of hMSCs are necessary for bone regeneration in tissue engineering. The present study aimed to design and develop a fluid flow mechanically-assisted cartridge device to enhance the osteogenic differentiation of hMSCs. We used the fluorescence-activated cell-sorting method to analyze the multipotent properties of hMSCs and found that the cultured cells retained their stemness potential. We also evaluated the cell viabilities of the cultured cells via water-soluble tetrazolium salt 1 (WST-1) assay under different rates of flow (0.035, 0.21, and 0.35 mL/min) and static conditions and found that the cell growth rate was approximately 12% higher in the 0.035 mL/min flow condition than the other conditions. Moreover, the cultured cells were healthy and adhered properly to the culture substrate. Enhanced mineralization and alkaline phosphatase activity were also observed under different perfusion conditions compared to the static conditions, indicating that the applied conditions play important roles in the proliferation and differentiation of hMSCs. Furthermore, we determined the expression levels of osteogenesis-related genes, including the runt-related protein 2 (Runx2), collagen type I (Col1), osteopontin (OPN), and osteocalcin (OCN), under various perfusion vis-à-vis static conditions and found that they were significantly affected by the applied conditions. Furthermore, the fluorescence intensities of OCN and OPN osteogenic gene markers were found to be enhanced in the 0.035 mL/min flow condition compared to the control, indicating that it was a suitable condition for osteogenic differentiation. Taken together, the findings of this study reveal that the developed cartridge device promotes the proliferation and differentiation of hMSCs and can potentially be used in the field of tissue engineering.

## 1. Introduction

The incidence of bone-related diseases, such as infections, osteoporosis, arthritis, and spinal disorders, is increasing due to the complex lifestyles of people. Autologous bone grafts are commonly used to treat major structural defects; however, their applications are restricted owing to donor-site morbidity and limited availability of graft materials [1]. Allotransplantion has also been applied to treat various damaged organs; however, the rejection of allografts or transmission of pathogens via these grafts have been reported in some cases [2]. Therefore, effective alternative approaches are required to treat such bone-related diseases. Tissue engineering is considered to be an alternative approach for the treatment and replacement of damaged organs, in which cells are cultivated outside the body and then transplanted into the host to regenerate the desired tissues or organs [3]. Despite several advantages of this method, heterogeneous cell death occurs at the center of thick, voluminous grafts in the static culture microenvironment due to the restriction of passive transport. Passive transport is limited to a short distance (100–200 μm), leading to the inappropriate supply and transport of nutrients, gases, metabolites, wastes, and, consequently, decreasing the cell viability [4,5,6,7]. Additionally, the native condition of the static system does not resemble that of most tissues. Microenvironments play important roles in the proliferation and differentiation of cells. It has been observed that mesenchymal stem cells (MSCs) facilitated osteoclastogenesis by producing the osteoclastogenic cytokines, macrophage colony-stimulating factor (M-CSF), and receptor activator for nuclear factor kappa B ligand (RANKL) under physiological conditions. However, MSCs inhibited the osteoclastogenesis by the secretion of the anti-osteoclastogenic factor osteoprotegerin (OPG) during inflammation [8]. The co-culture of MSCs with nucleus pulposus (NP) cells obtained from degenerated or non-degenerated human NP tissue indicated that the direct cellular interactions among MSCs and degenerated NPs cells triggered the differentiation of MSCs into NP-like phenotype and increased the endogenous NP cell populations [9]. A decrease in cancer tumorigenicity has been earlier reported in the co-culture of MSCs with ovarian cancer cells [10]. Cancer-associated MSCs (CA-MSCs) trigger tumor growth and angiogenesis by directly interacting with tumor cells or releasing cytokines, growth factors, and exosomes. The loss in the osteogenic and adipogenic differentiation potentials of MSCs has also been reported cultured within ovarian cancer cells, and co-cultured cells demonstrated CA-MSCs properties [11]. Other factors, such as pH, shear stress, and dissolved gas also influence the proliferation and differentiation of cells [12]. 

Different techniques, including the use of bioreactors, have been applied to generate an optimal cellular microenvironment. It is possible to control the different physicochemical factors, such as temperature, pH, humidity, pressure, shear stress, dissolved oxygen, and carbon dioxide, in the bioreactor [13,14,15]. The perfused media facilitate the continuous supply of nutrients as well as the removal of cellular wastes. Thus, bioreactors are considered as attractive and effective tools for use in tissue-engineering applications [16,17,18]. Bioreactors are utilized to culture different cells, including stem cells, osteoblasts, keratinocytes, cardiomyocytes, and myofibroblasts, for various applications [19]. Mechanostimulation (shear stress) is known to influence the maturation of cells by increasing the transportation of nutrients and waste [4,20,21,22,23]. Mechanostimulation has also been reported to enhance the construction of the extracellular matrix (ECM) as well as the secretion of cytokines and growth factors [24,25]. Furthermore, oxygen levels and shear stress, which triggers the expression of biological signals, play vital roles in cell proliferation and differentiation. Improved proliferation and differentiation of stem cells have been reported more under low oxygen conditions (2–5% O_2_) than at normal oxygen levels. Therefore, controlling the shear stress and oxygen levels is important for achieving better cellular activity [15,26]. 

The present study reported the proliferation and osteogenic differentiation potential of human mesenchymal stem cells (hMSCs) under fluid flow and static conditions in a novel cartridge device for tissue engineering. Several approaches have been utilized for the proliferation and osteogenic differentiation of hMSCs; however, the suitability of these approaches for tissue engineering remains unclear. An enhancement in cell viability, mineralization, and alkaline phosphatase activity was observed in the developed cartridge device at the appropriate shear stress (1.018 × 10^−6^ Pa). Upregulation of osteogenic transcription factors occurred more under fluid flow conditions than the static conditions in the developed device. Therefore, mechanical stimulation plays significant roles in the proliferation and osteogenic differentiation of hMSCs, and the developed device may be applied to promote the regeneration of bone tissues. 

## 2. Experiment Section 

### 2.1. Materials

Sodium phosphate, paraformaldehyde, alizarin red S (ARS), alcian blue, oil red O dye, Nissl stain (all purchased from Sigma-Aldrich, Burlington, MA, USA), bovine serum albumin (BSA) (ICN Biomedicals, Aurora, OH, USA), antibodies (BD Biosciences, San Jose, CA, USA), osteogenic, chondrogenic, adipogenic, and neurogenic differentiation media (Gibco BRL), water-soluble tetrazolium salt 1 (WST-1) assay (EZ-Cytox Cell Viability Assay Kit, Daeillab Service Co., LTD., Cheongwon-gun, Korea), ALP kit (AnaApec, Fremont, CA, USA), TRIzol reagent (Thermo Fischer Scientific, Waltham, MA, USA), reverse transcriptase (Superscript II RTase, Invitrogen, Gaithersburg, MD, USA), and SYBR Green Master Mix (Bio-Rad, Hercule, CA, USA) were used as received from the supplier without further purification. 

### 2.2. Development of Bioreactor Device

The fabricated system was composed of a mini chamber device (40 × 40 × 30 mm^3^) with inlets and outlets for the transport of media. A precision flowmeter was applied to the photographs of the developed system Figure 1a–d. The culture media was passed through a Tygon R3603 tube whose inner and outer diameters were 1.59 and 3.20 mm, respectively. A peristaltic pump (400 low-flow pump; Watson-Marlow Bredel Pump, Republic of Korea) was used to flow the cultured media. The fluid flow rate was adjusted to 0.035, 0.21, and 0.35 mL/min to achieve the shear stress of 1.018 × 10^−6^, 6.018 × 10^−6^, and 1.018 × 10^−5^ Pa, respectively. Static conditions were considered as the controls. The advantages of using the Tygon R3603 tube include its autoclavable and gas-permeable potential. An autoclave steam sterilizer was also used to sterilize the system (DAIHAN-Sci., Wonju, Republic of Korea). The rubber band (O-ring) and others were sterilized with 70% ethanol, followed by treatment with ultraviolet (UV) light. Different components of the developed system were assembled in a cell culture hood to avoid contamination. 

### 2.3. Cell Culture

The hMSCs were collected from the Intellectual Biointerface Engineering Center, Dental Research Institute, College of Dentistry, Seoul National University, Seoul, Republic of Korea. The cells were then treated with an α-minimum essential medium containing 10% fetal bovine serum (FBS) (Welgene Inc., Gyeongsan-si, Korea), 10 mM ascorbic acid (L-ascorbic acid), antibiotics, and sodium bicarbonate at 37 °C in a humidified atmosphere of 5% carbon dioxide (CO_2_). The old media was replaced with fresh media every day. After confluency was reached, the cells were separated with 1 mL trypsin-ethylenediaminetetraacetic acid (trypsin-EDTA) solution, followed by counting and passaging. Five passage cells were used in this study.

### 2.4. Flow Cytometric Analysis

Flow cytometric analysis was performed to examine the stemness potential of the cultured hMSCs. In brief, 1.0 × 10^4^ cells were fixed with a 3.7% paraformaldehyde (PFA) solution for 10 min, followed by resuspension in phosphate-buffered saline (PBS) with 1% BSA for 30 min to block the non-targeted antibody binding sites. The cells were then treated with specific antibodies against cluster of differentiation (CD)-34, CD13, CD90, or CD146 at 4 °C for 60 min, followed by treatment with fluorescent secondary antibodies at room temperature for 60 min. A FACSCalibur flow cytometer (Becton Dickinson Immunocytometry Systems, CA, USA) was used to analyze the percentages of CD34-negative and CD13-, CD90-, and CD-146-positive cells. 

### 2.5. Multi-Differentiation Potential of hMSCs 

The multi-differentiation potential of hMSCs was monitored in different differentiation media as previously reported [14]. Briefly, the cells were cultured in osteogenic, chondrogenic, adipogenic, and neurogenic differentiation media for the desired periods to assess the osteogenic, chondrogenic, adipogenic, and neurogenic abilities of hMSCs, respectively. The cells were then treated with 2% ARS stain (pH 4.2), 1% Alcian blue, 0.3% Oil Red O dye, and Nissl stain to examine the deposited calcified matrix, proteoglycans, fat vacuoles, and Nissl bodies as markers of osteogenic, chondrogenic, adipogenic, and neurogenic differentiation, respectively. An inverted light microscope (Olympus U-SPT; Olympus) was used to capture the images of the stained cells. 

### 2.6. Cell Viability and Mineralization

The WST-1 assay was used to analyze the viabilities of hMSCs in the developed systems after 3 days of cultivation under different shear stress conditions. The groups without shear stress were considered to be static (control). The cells were cultured for the desired periods and treated with WST-1 dye to form soluble formazan. A spectrophotometer (Victor 3; Perkin Elmer, Waltham, MA, USA) was then used to measure the absorbance of the formazan produced at 460 nm. 

The mineralization potentials of hMSCs were measured via ARS staining after 7 and 21 days of treatment under different shear conditions. The groups without shear stress were used as controls. In brief, 4 × 10^6^ cells were seeded on the surface of the plate and cultured for the chosen periods. After that, the cells were washed with PBS and fixed with 4% PFA solution at room temperature for 20 min, washed with distilled water, and stained with ARS media. After 30 min, the stained cells were rinsed with distilled water, and images were taken using an optical microscope. Quantification of the formed mineral was performed by treating the stained cells with the de-staining solution containing 10% cetylpyridinium chloride and 10 nM sodium phosphate, and absorbance was measured at 562 nm. All experiments were performed in triplicate (*n* = 3), and the data are presented as the mean optical density (OD) ± standard deviation (SD). Statistical significance was set at ** p* < 0.05.

### 2.7. Alkaline Phosphatase (ALP) Activity and Osteogenic Differentiation

The ALP activities of hMSCs were measured using a Sensolyte^TM^ ALP kit (AnaApec, USA) according to the manufacturer’s guidelines. The obtained cell suspension was centrifuged at 2500× *g* at 4 °C for 10 min, followed by the measurement of the absorbance of the yellow p-nitrophenol product at 405 nm with a spectrophotometer. ** p* < 0.05 was considered to be a significant difference.

The expression levels of the osteogenesis-related genes in hMSCs were determined via real-time polymerase chain reaction (RT-PCR) after 14 days of incubation. For this, 4 × 10^6^ cells were cultured in the developed bioreactor under different shear stress conditions for the desired periods. The groups without shear stress were used as controls. RNA extraction was performed using the TRIzol reagent according to the manufacturer’s guidelines. Before cDNA synthesis, the purity and concentration of the extracted RNA were determined using a spectrophotometer. cDNA was synthesized using reverse transcriptase and SYBR Green Master Mix. The Bio-Rad Real-Time PCR (CFX96^TM^ Maestro Real-Time Systems; Bio-Rad, Hercules, CA, USA) instrument was used to determine the mRNA-expression levels. This study involved 43 cycles of denaturation at 95 °C for 15 s and amplification at 60 °C for 1 min. All experiments were performed in triplicate (*n* = 3) and normalized to the housekeeping gene, hypoxanthine-guanine phosphoribosyl transferase (HPRT). The primer sets used in this study are listed in Table 1. For fluorescence staining, 4 × 10^4^ cells were fixed with 4% paraformaldehyde and permeabilized with 0.1% Triton-X 100, followed by blocking with 1% BSA for 60 min. After that, the cells were incubated with mouse anti-human OPN antibody (Santa Cruz Biotech, Santa Cruz, CA; USA) for 60 min, followed by incubation with AF/555 conjugated secondary antibody (Thermo Fischer Scientific, Waltham, MA, USA) for 60 min. Next, the cytoskeleton was stained with AF/488 conjugated F-actin probe (Thermo Fischer Scientific, USA) for 30 min. The cells were washed with PBS, and nuclei were stained with DAPI (Sigma-Aldrich, USA). The cells were mounted with Prolong Antifade mounting media, and cells were visualized with confocal microscope (LSM Zeiss,Zeiss, Oberkochen,Germany). The osteogenic differentiation medium was used for the mineralization, ALP activity, and osteogenic gene-expression experiments. 

### 2.8. Statistical Analysis

Statistical analysis was performed with one-way analysis of variance (ANOVA) using OriginPro v.9.0 (OriginLab, Northampton, MA, USA). All results are presented as the mean ± standard deviation. Statistical analyses were performed for the control and treated groups. Statistical significance was set at ** p* < 0.05, ** *p* < 0.01, and *** *p* < 0.001.

## 3. Results and Discussion

### 3.1. Bioreactor Setup

The developed system was designed using the computer-aided design (CAD) software, and the drawing views are shown in Figure 2. The side sketches of the designed system are shown in Figure 2a,b. The highlighted sections of the sketch indicate the flow of the culture media, and the direction of the culture media is from top to bottom. The top drawing views of the designed system with cover are shown in Figure 2c,d. The dimensions of the cover were 40 mm × 40 mm × 30 mm^3^. The developed system was connected to inlet and outlet ports associated with an external peristaltic pump to achieve proper mechanical stimulation. The streamline flow and velocity magnitude (m/s) of the cultured media in the developed system are shown in Figure 2e,f. Computational fluid dynamics (CFD) was used to calculate the shear stress that occurred on cells through the culture media. In this study, three flow intensities, 0.035, 0.21, and 0.35 mL/min, were considered, for which the shear stress value was 1.018 × 10^−6^, 6.108 × 10^−6^, and 1.018 × 10^−5^ Pa, respectively. The velocity magnitude was maximum at the inlet and outlet sections of the system and minimum at the center of the developed system. 

### 3.2. Stemness Potential of Cultured hMSCs

The stemness ability of the cultured hMSCs was analyzed using the fluorescence-activated cell-sorting (FACS) method, and the results are presented in Figure 3a. The cells cultured without perfusion conditions were taken for the stemness analysis. The cultured cells expressed enhanced (>90%) stem-cell-associated CD13, CD90, and CD146 markers and a decreased appearance of CD34 (~12%), demonstrating their stemness ability. Multi-differentiation and self-renewal are important characteristics of stem cells [27]. The loss of stemness ability of stem cells restricts its broad applicability. The multi-lineage differentiation ability of cultured hMSCs at different periods is shown in Figure 3b. The appearance of mineralized nodules, proteoglycans, fat vacuoles, and Nissl bodies after 7 days of culture confirmed the osteogenic, chondrogenic, adipogenic, and neurogenic differentiation of hMSCs, whose intensity increased after 14 and 21 days, showing the enhanced multi-lineage differentiation ability of hMSCs. Mesenchymal stem cells are commonly utilized in tissue engineering and regenerative medicine owing to their multi-lineage differentiation ability, high expansion, and good availability [28]. The stemness and multi-lineage differentiation abilities of stem cells are profoundly influenced by the surrounding microenvironment, cell–cell interactions, and others [29]. 

### 3.3. Cell Viability and Morphology 

To validate the design of the developed system, we performed a comparative cell-viability study between the conventional culture system (5% CO_2_ incubator) and the developed bioreactor after 3 days of treatment, and the results are given in Appendix A. The cell-growth rate was similar in both conditions, indicating that the developed bioreactor has no adverse effects on hMSCs and is optimized. The viability of the cultured hMSCs was measured by WST-1 assay after 3 days of treatment in the developed system under different shear stress conditions, and the results are shown in Figure 4a. Enhanced cell viability was observed under shear stress conditions compared to the static condition, and this was a more significant inflow rate of 0.035 mL/min (1.018 × 10^−6^ Pa) than the other stress conditions, suggesting the ideal stress for cellular activity. The enhancement in cell viability under shear stress conditions is attributed to the enhancement of collagen deposition, nutrient transport, and uptake, facilitating the upregulation of cellular activities. Shear stress plays a significant role in the cell shape and intracellular signaling pathways. Shear stress-induced alignment of different types of cells has been previously reported via the reorganization of the cytoskeleton [30,31]. Furthermore, decreased cell viability occurred with increasing magnitude of shear stress due to stress-induced cell death. Therefore, it is important to maintain stress levels to improve cellular activity. Additionally, the microstructures of the material used for the cell culture also influence cellular activity [32]. 

The morphologies of the cultured cells were examined using a light microscope, and the morphologies are shown in Figure 4b. The cultured cells were adequately attached to the substrate and elongated in shape. The cells were healthy and covered the entire culture surface. The cell densities were higher under shear stress conditions compared to the static condition, and this was higher in the 1.018 × 10^−6^ Pa condition than in the others, indicating a suitable condition for cell growth, as observed in the cell viability results. 

### 3.4. Effects of Shear Stress on Mineralization and ALP Activity 

Bone tissues are enriched with calcium content, and the detection of the deposited minerals in the osteogenic media provides essential information about cellular activity. The deposited minerals are the primary indication of osteogenesis [33]. The optical microscopic images of the ARS stain-treated hMSCs under different shear stress conditions after 7 and 21 days of culture are shown in Figure 5a. The groups without shear stress treatment were used as controls. The shear stress-treated groups exhibited more red spots than the control after 7 d of culture, showing a positive effect on mineralization. The intensities of the generated red spots were further enhanced in the stress-treated groups after 21 days of treatment, demonstrating improved mineralization ability. Among these (1.018 × 10^−6^, 6.108 × 10^−6^, and 1.018 × 10^−5^ Pa), the 1.018 × 10^−6^ Pa stress-treated groups exhibited greater mineralization potential than the others due to the better cellular activity conditions. The quantitative values of the formed nodules were measured using a spectrophotometer after 21 d of incubation, and the values are presented in Figure 5b. The mineralized values were higher in the shear stress-treated groups than in the control, indicating the positive effects of shear stress on the mineralization of hMSCs. This value was more prominent in the 1.018 × 10^−6^ Pa stress-treated groups than in the other groups because of the better cellular activity at this shear stress. 

The ALP activity of cultured hMSCs was also measured under different shear conditions after 14 d of incubation, and the results are presented in Figure 5c. The groups without any shear stress were considered as the control (static). ALP activity also provides useful information on osteogenesis. ALP activity was prominently upregulated in shear stress-mediated groups compared to the control, showing positive effects of shear stress on osteogenesis of hMSCs. This was more up-regulated in the 1.018 × 10^−6^ Pa stress-treated groups than in the other groups. ALP is an ectoenzyme that triggers mineralization by involving the degradation of inorganic pyrophosphates. These degraded inorganic phosphates act as templates for mineralization [34]. 

### 3.5. Shear Stress-Induced Osteogenic Differentiation of hMSCs 

The osteogenic differentiation capabilities of hMSCs cultured in the developed system under different shear stress conditions were analyzed by RT-PCR, and the results are presented in Figure 6a–d. The improved expression levels of the osteogenesis-related genes (runt-related protein 2 (Runx2), collagen type I (Col1), osteopontin (OPN), and osteocalcin (OCN)) in hMSCs were observed under different shear stress conditions compared to the control, indicating the accelerating effects of shear stress on osteogenic differentiation. It was more significant in the 1.018 × 10^−6^ Pa stress-treated groups than the other groups due to the favorable conditions for enhanced cellular activities. *Runx2* is an important osteogenic gene marker, and it is expressed at an early stage of cell differentiation. It plays a crucial role in osteogenic differentiation. Col1 is an early osteogenic transcription factor, and its expression indicates the presence of bone cells [35,36,37,38]. *OCN* is considered one of the most important late osteogenesis-associated gene markers, which are found during mineralization and assembled in mineralized bones. OPN is also known as secreted phosphoprotein 1 (SPP1), and its expression is upregulated during osteogenic differentiation. Different factors, such as stimuli, cultured microenvironment, and the presence of scaffolds/nanomaterials, directly influence the osteogenic potentials of the stem cells [39,40]. Mechanical stimuli are classified into material properties, ECM, and different stresses. The appropriate mechanical stimuli can alter the properties of ECM, leading to improved bone growth and regeneration. The change in cell morphology, motility, and function was previously reported with mechanical stimulation due to the stretching of the actin cytoskeleton [41,42]. Enhanced expression of myogenic factors was observed with low-frequency periodic mechanical stimulation in C2C12 myoblasts by increasing the microRNA content and facilitating the myogenic differentiation. It was observed that the mineralization was significantly higher compared to the expression of osteogenic-related markers under the shear stress conditions. This can be attributed to the improved cellular activity, which stimulated the matrix vesicle (MV)-assisted mineralization, followed by collagen-assisted mineralization. The MV-mediated mineralization led to the formation of the apatite moieties within the matrix vesicles and transportation outside the cells. This preformed apatite deposited into the collagenous extracellular matrix (ECM), causing the mineralization through different processes [43]. The enhanced cell proliferation under the shear-stress conditions facilitates greater mineralization [14]. It has been observed that beta-catenin plays an important role in the flow-induced osteogenic differentiation of MSCs [44].

The morphologies of hMSCs cultured in the developed systems as determined by confocal laser scanning microscopy are presented in Figure 7. Here, we chose the 1.018 × 10^−6^ Pa shear stress to express OPN marker due to the improved osteogenic differentiation under these conditions. The groups without any stress were used as controls. Shear stress-treated cells exhibited higher expression levels of OPN gene marker compared to the control. Moreover, the cells were properly aligned and adhered to the surfaces. The limitation of our study is that the approach of mechano-transduction in hMSCs used in this study was not explored under different shear stress conditions. However, we demonstrated that shear stress plays important roles in the proliferation and differentiation of cells. We believe that the nutrients and surrounding ECM changes might be crucial factors that affect the shear stress in cultured cells. 

## 4. Conclusions

The bioreactor is a vital tool for culturing various types of cells for applications in tissue engineering. In this study, we designed and developed a perfusion flow bioreactor system and evaluated its tissue-engineering potential. We found that the cells cultured under the developed bioreactor system without perfusion exhibited a similar trend of cell growth as the conventional cell-culture system (5% CO_2_ incubator). No significant differences in the cellular morphology were observed in the developed bioreactor (with perfusions), and static condition indicated the potential of the designed and developed bioreactor. Furthermore, the cell viability rate was significantly higher (12%) under perfusion conditions than static conditions, which was significantly affected by shear stress. We also found that lower shear stress conditions facilitated increased cell growth, mineralization, and differentiation of osteocytes compared to high-stress conditions. Moreover, the morphologies of the cultured cells can be monitored and captured using a microscope. Therefore, the perfusion bioreactor system designed and developed in this study can potentially be used for the rapid growth of cells in various tissue-engineering applications.

## Figures and Tables

**Figure 1 micromachines-12-00927-f001:**
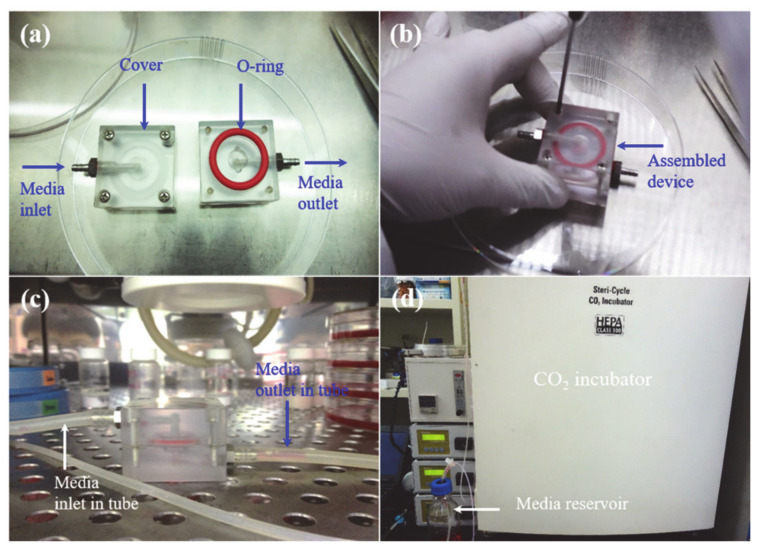
Photographs of the developed device at different conditions: (**a**) different parts of the developed device in open condition (**b**) in assembled condition (**c**) developed device in CO_2_ incubator connected with tube (**d**) images of used CO_2_ incubator with media reservoir and other controllers.

**Figure 2 micromachines-12-00927-f002:**
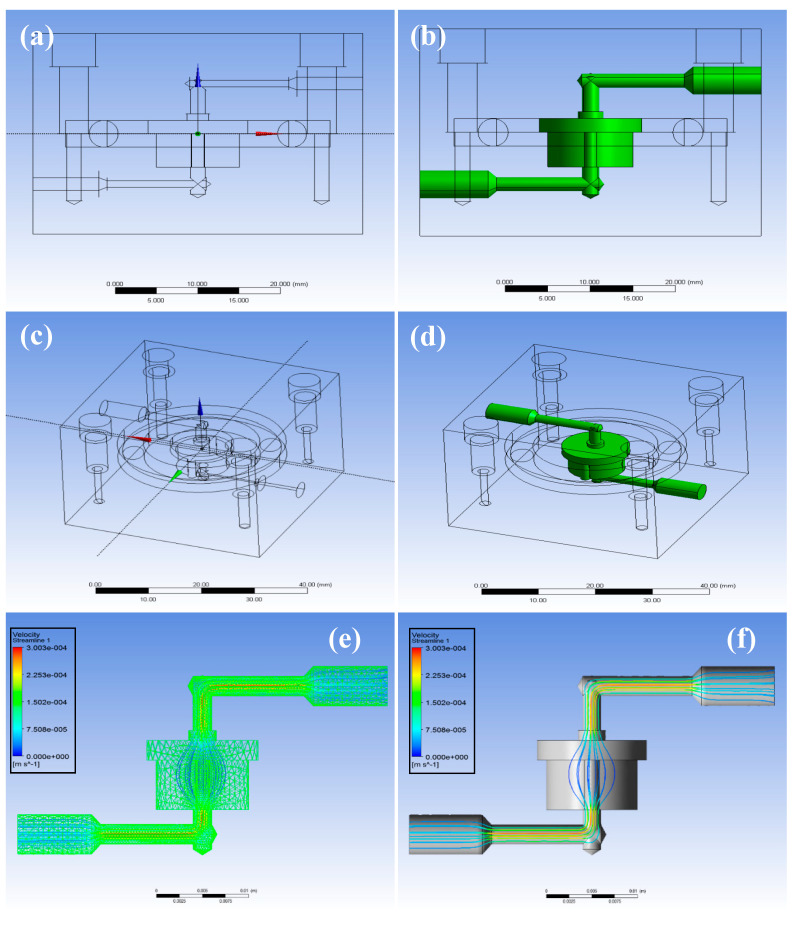
Computer-Aided Design (CAD) design of the developed device: (**a**,**b**) side drawing views (**c**,**d**) top drawing views with covers and (**e**,**f**) velocity magnitude (m/s) and streamlines for the media flow.

**Figure 3 micromachines-12-00927-f003:**
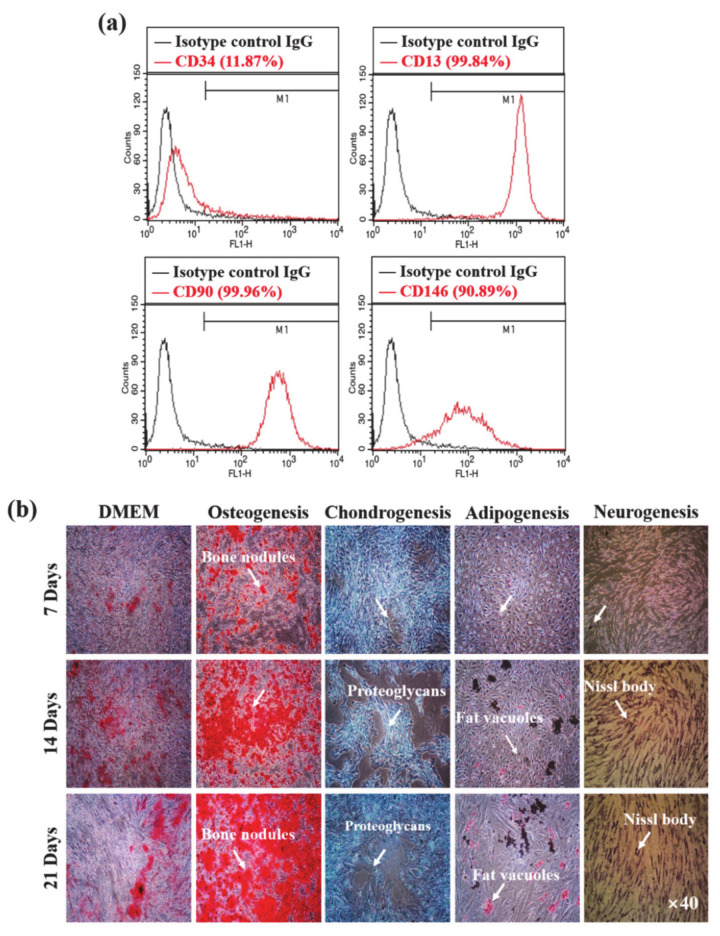
Stemness potential analysis of Human mesenchymal stem cells (hMSCs) in the developed device (**a**) The cell surface antigen profile of hMSCs was confirmed by flow cytometry (**b**) The multi-differentiation potential of hMSCs grown in the developed device.

**Figure 4 micromachines-12-00927-f004:**
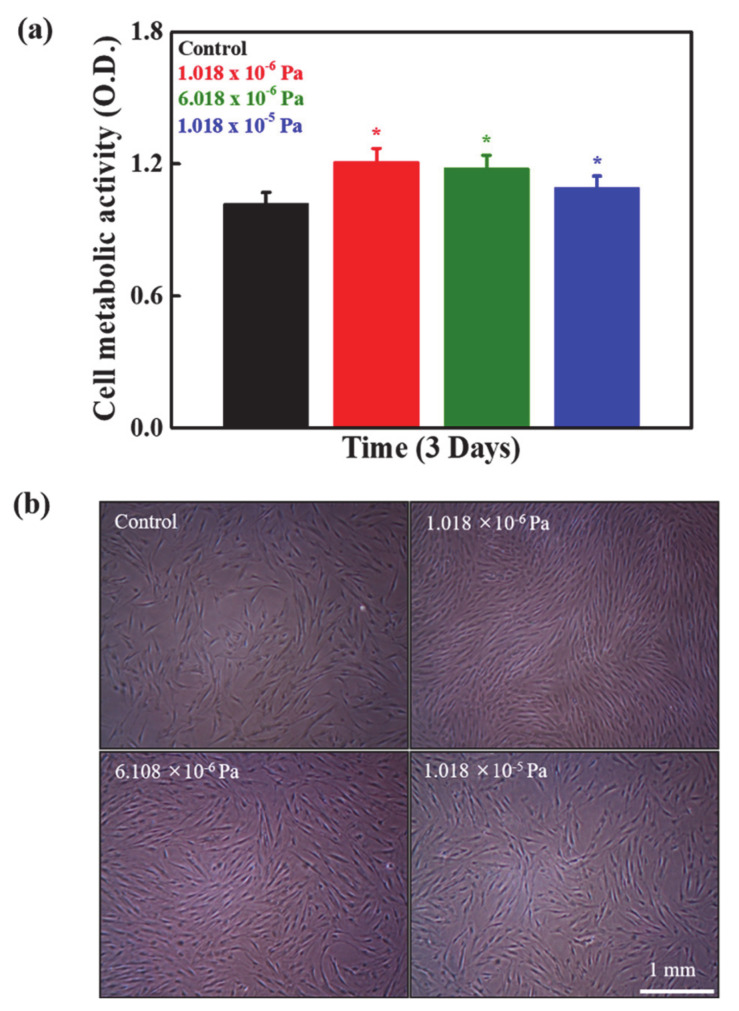
(**a**) Viability of the cultured hMSCs in the developed device at different shear stresses after three days interval Statistical analyses were performed for the control and treated groups. Statistical significance was set at ** p* < 0.05, (**b**) the morphologies of the cultured hMSCs in the developed device at indicated conditions.

**Figure 5 micromachines-12-00927-f005:**
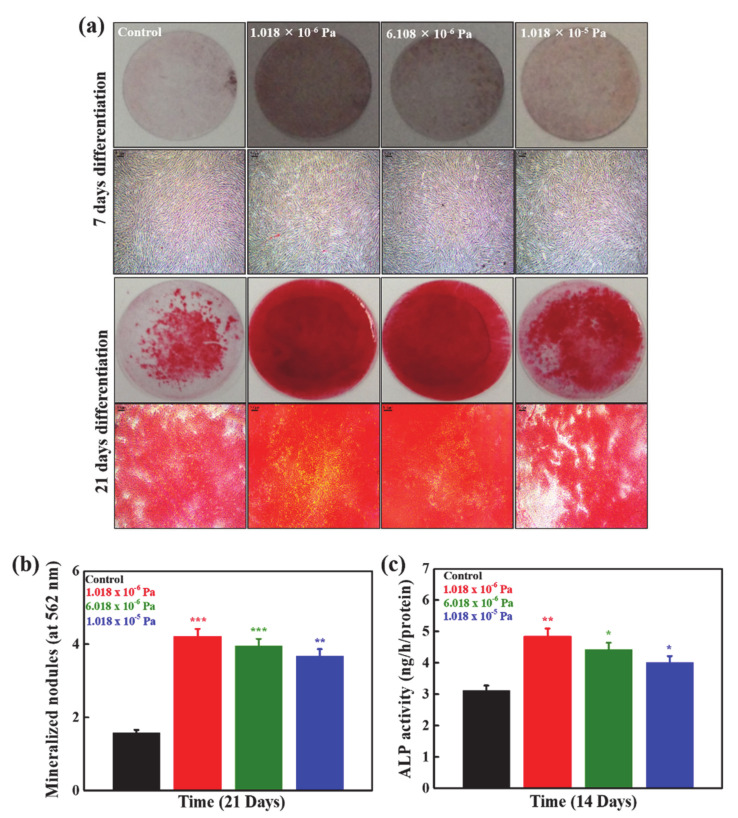
(**a**) The qualitative analysis of the mineralization potential of hMSCs in the developed device under different shear stress conditions at indicated time periods (**b**) the quantitative values of the formed minerals in the developed device under different shear stress conditions after 21 days of time periods, and (**c**) the ALP activity of hMSCs in the developed device after 14 days of incubation. Statistical analyses were performed for the control and treated groups. Statistical significance was set at ** p* < 0.05, ** *p* < 0.01, and *** *p* < 0.001.

**Figure 6 micromachines-12-00927-f006:**
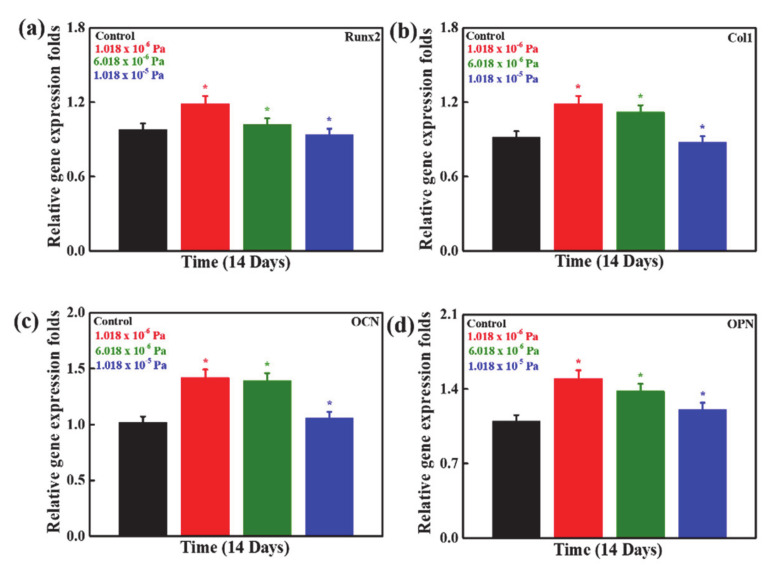
The expression of osteogenic associated gene markers in hMSCs in the developed device after 14 days of cultivation (**a**) the relative expression of Col1 (**b**) Col1 (**c**) OCN (**d**) OPN after 14 days of cultivation. Statistical analyses were performed for the control and treated groups. Statistical significance was set at ** p* < 0.05.

**Figure 7 micromachines-12-00927-f007:**
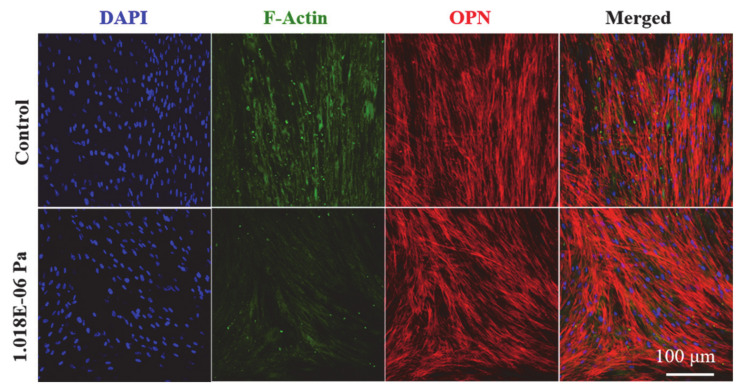
ICC results. The expression of osteogenic-related gene marker at indicated conditions in the developed device.

**Table 1 micromachines-12-00927-t001:** The primer sequences applied in this study.

Gene	Accession No.	Sequences
HPRT	NM_000194	5′-GCGCAAGTACTCTGTGTGGA-3′ 5′-ACATCTGCTGGAAGGTGGAC-3′
Runx2	NM_001146038	5′-GGACATGCAGTACGAGCTGA-3′ 5′-GCAGTGAAGGGCTTCTTGTC-3′
Col1	NM007742	5′-TGACCTTCCTGCGCCTGATGTCC-3′ 5′-CTGGGGCACCAACGTCCAAGGG-3′
OCN	AL135927	5′-GTGCAGAGTCCAGCAAAGGT-3′ 5′-TCAGCCAACTCGTCACAGTC-3′
OPN	J04765	5′-GAAACGAGTCAGCTGGATG-3′ 5′-TGAAATTCATGGCTGTGGAA-3′

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
