# Peer review of "Fluid Flow Mechanical Stimulation-Assisted Cartridge Device for the Osteogenic Differentiation of Human Mesenchymal Stem Cells"

_micromachines, 2021, doi:10.3390/mi12080927_

Round 1

Reviewer 1 Report

The manuscript by Ki-Taek Lim et al reports on an innovative device for the osteogenic differentiation of mesenchymal cells. The idea is interesting, however several major comments should be addressed: Some examples regarding the applicability of the models should be provided:

  • what happens if the hMSCs are co-cultured with other normal cells from bone marrow (Osteoclasts/myeloid cells)?
  • what if cancer cells are present at the same time? could they affect osteoblastic differentiation?
  • Have the authors tried to standardize the model by using immortalized cell lines such as HS5? please take into account that MSC in general could be very sensitive to the quality of the material used for the construction of the chamber (Limongi T, et al Influence of the Fabrication Accuracy of Hot-Embossed PCL Scaffolds on Cell Growths. Front Bioeng Biotechnol. 2020 Feb 14;8:84. doi: 10.3389/fbioe.2020.00084)

Author Response

Comments and Suggestions for Authors

The manuscript by Ki-Taek Lim et al reports on an innovative device for the osteogenic differentiation of mesenchymal cells. The idea is interesting, however several major comments should be addressed: Some examples regarding the applicability of the models should be provided:

Reply to the reviewer: Thank you very much for reviewing the manuscript. We appreciate the reviewer's suggestions to improve the quality of this study.

  1. What happens if the hMSCs are co-cultured with other normal cells from bone marrow (Osteoclasts/myeloid cells)?

Response: Thank you very much for reviewing the manuscript. We have added the effects of MSCs on other cells in the revised manuscript.

It has been observed that mesenchymal stem cells (MSCs) facilitated osteoclastogenesis by producing the osteoclastogenic cytokines, macrophage colony-stimulating factor (M-CSF), and receptor activator for nuclear factor kappa B ligand (RANKL) under physiological conditions. However, MSCs inhibited the osteoclastogenesis by the secretion of the anti-osteoclastogenic factor, osteoprotegerin (OPG), during inflammation [8]. The co-culture of MSCs with nucleus pulposus (NP) cells obtained from degenerated or non-degenerated human NP tissue indicated that the direct cellular interactions among MSCs and degenerated NPs cells triggered the differentiation of MSCs into NP-like phenotype and increased the endogenous NP cell populations [9].

  1. What if cancer cells are present at the same time? Could they affect osteoblastic differentiation?

Response: Thank you very much for reviewing the manuscript. We have described the effects of cancer cells on MSCs in the revised manuscript.

A decrease in cancer tumorigenicity has been earlier reported in the co-culture of MSCs with ovarian cancer cells [10]. Cancer-associated MSCs (CA-MSCs) trigger tumor growth, and angiogenesis by directly interacting with tumor cells or releasing cytokines, growth factors or exosomes.

The loss in the osteogenic and adipogenic differentiation potentials of MSCs has also been reported cultured within ovarian cancer cells, and co-cultured cells demonstrated CA-MSCs properties [11].

  1. Have the authors tried to standardize the model by using immortalized cell lines such as HS5? Please take into account that MSC in general could be very sensitive to the quality of the material used for the construction of the chamber (Limongi T, et al Influence of the Fabrication Accuracy of Hot-Embossed PCL Scaffolds on Cell Growths. Front Bioeng Biotechnol. 2020 Feb 14;8:84. doi: 10.3389/fbioe.2020.00084).

Response: Thank you very much for reviewing the manuscript. We have standardized the developed device by a comparative cell viability study between the conventional culture system (5% CO2 incubator), and the developed bioreactor after 3 d of treatment, and the results are given in Figure S1. The cell growth rate was similar in both conditions, indicating that the developed bioreactor has no adverse effects on hMSCs and is optimized.

The different parameters, including the microstructures of the material used for the cell culture, have a vast influence on cellular activity [32].

Reviewer 2 Report

Ki-Taek Lim and colleagues developed a novel fluid flow mechanically-assisted device, and investigated its application in the osteogenic differentiation of human mesenchymal stem cells (hMSCs). They showed that the stemness of hMSCs was preserved using the developed device. By comparing different flow rate conditions, they found the 0.035 mL/min flow condition gave the highest cell viability, and enhanced osteogenic differentiation capacity of hMSCs. The experiment was properly designed, and the results were well presented and described, however, there are some concerns and suggestions:

Major concerns:

Figure. 5 showed a significant increased mineralization level of hMSCs under shear stress condition compared with no stress condition. However, the osteogenic associated gene was only slightly increased compared with control group (Figure. 6). Have the authors measured the cell proliferation rate at longer culture period, such as 7, 14 or 21 days. Is the significant increased mineralization level mainly due to a higher proliferation rate in the shear stress conditions?

It has been reported that steady fluid flow can activate beta-catenin, which is necessary for flow induced osteogenic differentiation of MSCs (Emily J. Arnsdorf, et al. PloS one 2009, 4 (4), e5388). Besides the genes shown in Figure. 6, have the authors also measured other osteogenic associated gens, such as beta-catenin?

Minor concerns:

Please clarify the definition of “control cells” in line 333. Please also clarify the differences between “without perfusion” and “static conditions” in line 334.

Is the “3.2 Stemness potential of cultured hMSCs” part done without shear stress (static)? If it is the case, please describe the condition in the results part.

There seems a typo in the legend of Figure. 3b. Please remove "3.3 Cell viability and morphology".

There are two periods at the last sentence of the legends of Figure. 2, Figure. 4 and Figure. 5.

Please clarify the culture medium that was used for Figure. 5, Figure. 6 and Figure. 7. Are all these experiments done with osteogenic differentiation medium?

Please change “Time (Days)” to “Times (14 Days)” for the x-axis of Figure. 6a-d.

Author Response

Comments and Suggestions for Authors

Ki-Taek Lim and colleagues developed a novel fluid flow mechanically-assisted device, and investigated its application in the osteogenic differentiation of human mesenchymal stem cells (hMSCs). They showed that the stemness of hMSCs was preserved using the developed device. By comparing different flow rate conditions, they found the 0.035 mL/min flow condition gave the highest cell viability, and enhanced osteogenic differentiation capacity of hMSCs. The experiment was properly designed, and the results were well presented and described, however, there are some concerns and suggestions:

Reply to the reviewer: Thank you very much for reviewing the manuscript. We appreciate the reviewer's suggestions to improve the quality of this study.

Major concerns:

  1. Figure. 5 showed a significant increased mineralization level of hMSCs under shear stress condition compared with no stress condition. However, the osteogenic associated gene was only slightly increased compared with control group (Figure. 6). Have the authors measured the cell proliferation rate at longer culture period, such as 7, 14 or 21 days. Is the significant increased mineralization level mainly due to a higher proliferation rate in the shear stress conditions?

Response: Thank you very much for reviewing the manuscript. We have measured the cell proliferation rate after 3 days of the treatment under different conditions in this work. Yes, the improved mineralization was due to a higher proliferation rate in the shear stress conditions and this observation is supported by suitable references.

It was observed that the mineralization was significantly higher compared to the expression of osteogenic-related markers under the shear stress conditions. This can be attributed to the improved cellular activity, which stimulated the matrix vesicle (MV)–assisted mineralization, followed by collagen-assisted mineralization. The MV–mediated mineralization led to the formation of the apatite moieties within the matrix vesicles and transported outside the cells. This preformed apatite deposited into the collagenous extracellular matrix (ECM), causing the mineralization through different processes [43]. The enhanced cell proliferation under the shear-stress conditions facilitates greater mineralization [14].

  1. It has been reported that steady fluid flow can activate beta-catenin, which is necessary for flow induced osteogenic differentiation of MSCs (Emily J. Arnsdorf, et al. PloS one 2009, 4 (4), e5388). Besides the genes shown in Figure. 6, have the authors also measured other osteogenic associated gens, such as beta-catenin?

Response: Thank you very much for reviewing the manuscript. Yes, we agree with you. The steady fluid flow activates beta-catenin, which plays an important role in osteogenic differentiation [44]. Here, we have evaluated the relative expression of other crucial osteogenic gene markers, including Runx2, Col1, OCN, and OPN instead of beta-catenin.

Minor concerns:

  1. Please clarify the definition of “control cells” in line 333. Please also clarify the differences between “without perfusion” and “static conditions” in line 334.

Response: Thank you very much for reviewing the manuscript. We have clarified it in the revised manuscript.

We found that the cells cultured under the developed bioreactor system without perfusion exhibited a similar trend of cell growth as the conventional cell culture system (5% CO2 incubator). No significant differences in the cellular morphology were observed in the developed bioreactor (with perfusions), and static conditions indicated the potential of the designed and developed bioreactor

  1. Is the “3.2 Stemness potential of cultured hMSCs” part done without shear stress (static)? If it is the case, please describe the condition in the results part.

Response: Thank you very much for reviewing the manuscript. We have described it in the revised manuscript.

The cells cultured without perfusion conditions were taken for the stemness analysis.  

  1. There seems a typo in the legend of Figure. 3b. Please remove "3.3 Cell viability and morphology".

Response: Thank you very much for your kind suggestion. We have corrected it in the revised manuscript.

  1. There are two periods at the last sentence of the legends of Figure. 2, Figure. 4 and Figure. 5.

Response: Thank you very much for your kind suggestion. We have corrected it in the revised manuscript.

  1. Please clarify the culture medium that was used for Figure. 5, Figure. 6 and Figure. 7. Are all these experiments done with osteogenic differentiation medium?

Response: Thank you very much for reviewing the manuscript. We have described the culture medium for the osteogenic differentiation in the revised manuscript.

The osteogenic differentiation medium was used for the mineralization, ALP activity, and osteogenic gene expression experiments.

  1. Please change “Time (Days)” to “Times (14 Days)” for the x-axis of Figure. 6a-d.

Response: Thank you very much for your kind suggestion. We have corrected it in the revised manuscript.

Round 2

Reviewer 1 Report

The authors satisfactory introduced potential uses of the model in other fields by improving the introduction and the conclusion.

Reviewer 2 Report

The authors has addressed all my concerns.